# PonderNet: Learning to Ponder

**Andrea Banino**[*]
**DeepMind**
**London, UK**
`abanino@deepmind.com`

**Jan Balaguer\***
**DeepMind**
**London, UK**
`jan@deepmind.com`

**Charles Blundell**
**DeepMind**
**London, UK**
`cblundell@deepmind.com`

## Abstract

In standard neural networks the amount of computation used grows with the size of the inputs, but not with the complexity of the problem being learnt. To overcome this limitation we introduce PonderNet, a new algorithm that learns to adapt the amount of computation based on the complexity of the problem at hand. PonderNet learns end-to-end the number of computational steps to achieve an effective compromise between training prediction accuracy, computational cost and generalization. On a complex synthetic problem, PonderNet dramatically improves performance over previous adaptive computation methods and additionally succeeds at extrapolation tests where traditional neural networks fail. Also, our method matched the current state of the art results on a real world question and answering dataset, but using less compute. Finally, PonderNet reached state of the art results on a complex task designed to test the reasoning capabilities of neural networks.

## 1. Introduction

The time required to solve a problem is a function of more than just the size of the inputs. Commonly problems also have an inherent complexity that is independent of the input size: it is faster to add two numbers than to divide them. Most machine learning algorithms do not adjust their computational budget based on the complexity of the task they are learning to solve, or arguably, such adaptation is done manually by the machine learning practitioner. This adaptation is known as pondering. In prior work, Adaptive Computation Time (ACT; Graves, 2016) automatically learns to scale the required computation time via a scalar halting probability. This halting probability modulates the number of computational steps, called the "ponder time", needed for each input. Unfortunately ACT is notably unstable and

---

*. contributed equally

sensitive to the choice of a hyper-parameter that trades-off accuracy and computation cost. Additionally, the gradient for the cost of computation can only back-propagate through the last computational step, leading to a biased estimation of the gradient. Another approach is represented by Adaptive Early Exit Networks (Bolukbasi et al., 2017) where the forward pass of an existing network is terminated at evaluation time if it is likely that the part of the network used so far already predicts the correct answer. More recently, work has investigated the use of REINFORCE (Williams, 1992) to perform conditional computation. A discrete latent variable is used to dynamically adjust the number of computation steps. This approach has been applied to recurrent neural networks (Chung et al., 2016; Banino et al., 2020), but has the downside that the estimated gradients have high variance, requiring large batch sizes to train them. A parallel line of research has explored using similar techniques to reduce the computation by skipping elements from a sequence of processed inputs (Yu et al., 2017; Campos Camunez et al., 2018).

In this paper we present PonderNet that builds on these previous ideas. PonderNet is fully differentiable which allows for low-variance gradient estimates (unlike REINFORCE). It has unbiased gradient estimates (unlike ACT). We achieve this by reformulating the halting policy as a probabilistic model. This has consequences in all aspects of the model:

1. Architecture: in PonderNet, the halting node predicts the probability of halting conditional on not having halted before. We exactly compute the overall probability of halting at each step as a geometric distribution.

2. Loss: we don't regularize PonderNet to explicitly minimize the number of computing steps, but incentivize exploration instead. The pressure of using computation efficiently happens naturally as a form of Occam's razor.

3. Inference: PonderNet is probabilistic both in terms of number of computational steps and the prediction produced by the network.

## 2. Methods

### 2.1 Problem setting

We consider a supervised setting, where we want to learn a function $f : x \to y$ from data $(\mathbf{x}, \mathbf{y})$, with $\mathbf{x} = \{x^{(1)}, ..., x^{(k)}\}$ and $\mathbf{y} = \{y^{(1)}, ..., y^{(k)}\}$. We propose a new general architecture for neural networks that modifies the forward pass, as well as a novel loss function to train it.

### 2.2 Step recurrence and halting process

The *PonderNet* architecture requires a *step function* $s$ of the form $\hat{y}_n, h_{n+1}, \lambda_n = s(x, h_n)$, as well as an initial state $h_0$ [1]. The output $\hat{y}_n$ and $\lambda_n$ are respectively the network's prediction and scalar probability of halting at step $n$. The step function $s$ can be any neural network, such as MLPs, LSTMs, or encoder-decoder architectures such as transformers. We apply the step function recurrently up to $N$ times.

---

1. Alternatively, one can consider a step function of the form $\hat{y}_n, h_{n+1}, \lambda_n = s(h_n)$ together with an *encoder* $e$ of the form $h_0 = e(x)$.

The output $\hat{y}_n$ is a learned prediction conditioned on the dynamic number of steps $n \in \{1, \ldots, N\}$. We rely on the value of $\lambda_n$ to learn the optimal value of $n$. We define a Bernoulli random variable $\Lambda_n$ in order to represent a Markov process for the halting with two states "continue" ($\Lambda_n = 0$) and "halt" ($\Lambda_n = 1$). The decision process starts from state "continue" ($\Lambda_0 = 0$). We set the transition probability:

$$P(\Lambda_n = 1 | \Lambda_{n-1} = 0) = \lambda_n \quad \forall \, 1 \leq n \leq N \tag{1}$$

that is the conditional probability of entering state "halt" at step $n$ conditioned that there has been no previous halting. Note that "halt" is a terminal state. We can then estimate the unconditioned probability that the halting happened in steps $0, 1, 2, ..., N$ where $N$ is the maximum number of steps allowed before halting. We derive this probability distribution $p_n$ as a generalization of the geometric distribution:

$$p_n = \lambda_n \prod_{j=1}^{n-1}(1 - \lambda_j) \tag{2}$$

which is a valid probability distribution if we integrate over an infinite number of possible computation steps ($N \to \infty$).

The prediction $\hat{y} \sim \hat{Y}$ made by PonderNet is sampled from a random variable $\hat{Y}$ with probability distribution $P(\hat{Y} = y_n) = p_n$. In other words, the prediction of PonderNet is the prediction made at the step $n$ at which it halts. This is in contrast with ACT, where model predictions are always weighted averages across steps. Additionally, PonderNet is more generic in this regard: if one wishes to do so, it is straightforward to calculate the expected prediction across steps, similar to how it is done in ACT.

## 2.3 Maximum number of pondering steps

Since in practice we can only unroll the step function for a limited number of iterations, we must correct for this so that the sum of probabilities $p_n$ sums to 1. We can do this in two ways. One option here is to normalize the probabilities $p_n$ so that they sum up to 1 (this is equivalent to conditioning the probability of halting under the knowledge that $n \leq N$). Alternatively, we could assign any remaining halting probability to the last step, so that $p_N = 1 - \sum_{n=1}^{N-1} p_n$ instead of as previously defined.

In our experiments, we specify the maximum number of steps using two different criteria. In evaluation, and under known temporal or computational limitations, $N$ can be set naively as a constant (or not set any limit, i.e. $N \to \infty$). For training, we found that a more effective (and interpretable) way of parameterizing $N$ is by defining a minimum cumulative probability of halting. $N$ is then the smallest value of $n$ such that $\sum_{j=1}^{n} p_j > 1 - \varepsilon$, with the hyper-parameter $\varepsilon$ positive near 0 (in our experiments 0.05).

## 2.4 Training loss

The total loss is composed of reconstruction $L_{Rec}$ and regularization $L_{Reg}$ terms:

$$L = \underbrace{\sum_{n=1}^{N} p_n \mathcal{L}(y, \hat{y}_n)}_{L_{Rec}} + \beta \underbrace{KL(p_n||p_G(\lambda_p))}_{L_{Reg}} \tag{3}$$

where $\mathcal{L}$ is a pre-defined loss for the prediction (usually mean squared error, or cross-entropy); and $\lambda_p$ is a hyper-parameter that defines a geometric prior distribution $p_G(\lambda_p)$ on the halting policy (truncated at N). $L_{Rec}$ is the expectation of the pre-defined reconstruction loss $\mathcal{L}$ across halting steps. $L_{Reg}$ is the KL divergence between the distribution of halting probabilities $p_n$ and the prior (a geometric distribution truncated at N, parameterized by $\lambda_p$). This hyper-parameter defines a prior on how likely it is that the network will halt at each step. This regularisation serves two purposes. First, it biases the network towards the expected prior number of steps $1/\lambda_p$. Second, it provides an incentive to give a non-zero probability to all possible number of steps, thus promoting exploration.

### 2.5 Evaluation sampling

At evaluation, the network samples on a step basis from the halting Bernoulli random variable $\Lambda_n \sim B(p = \lambda_n)$ to decide whether to continue or to halt. This process is repeated on every step $n$ until a "halt" outcome is sampled, at which point the output $y = y_n$ becomes the final prediction of the network. If a maximum number of steps $N$ is reached, the network is automatically halted and produces a prediction $y = y_N$.

## 3. Results

### 3.1 Parity

In this section we are reporting results on the parity task as introduced in the original ACT paper (Graves, 2016). Out of the four tasks presented in that paper we decided to focus on parity as it was the one showing greater benefit from adaptive compute. In our instantiation of the parity problem the input vectors had 64 elements, of which a random number from 1 to 64 were randomly set to 1 or $-1$ and the rest were set to 0. The corresponding target was 1 if there was an odd number of ones and 0 if there was an even number of ones. We refer the reader to the original ACT paper for specific details on the tasks (Graves, 2016). Also, please refer to Appendix B for further training and evaluation details.

In figure 1a we can see that PonderNet achieved better accuracy than ACT on the parity task and it did so with a more efficient use of thinking time (1a at the bottom). Moreover, if we consider the total computation time during training (figure 1c) we can see that, in comparison to ACT, PonderNet employed less computation and achieved higher score.

Another analysis we performed on this version of the parity task was to look at the effect of the prior probability on performance. In figure 2b we show that the only case where PonderNet could not solve the task is when the prior ($\lambda_p$) was set to 0.9, that is when the average number of thinking steps given as prior was roughly 1 (1/0.9). Interestingly, when the prior ($\lambda_p$) was set to 0.1, hence starting with a prior average thinking time of 10 steps (1/0.1), the network managed to overcome this and settled to a more efficient average thinking time of roughly 3 steps (figure 2c). These results are important as they show that

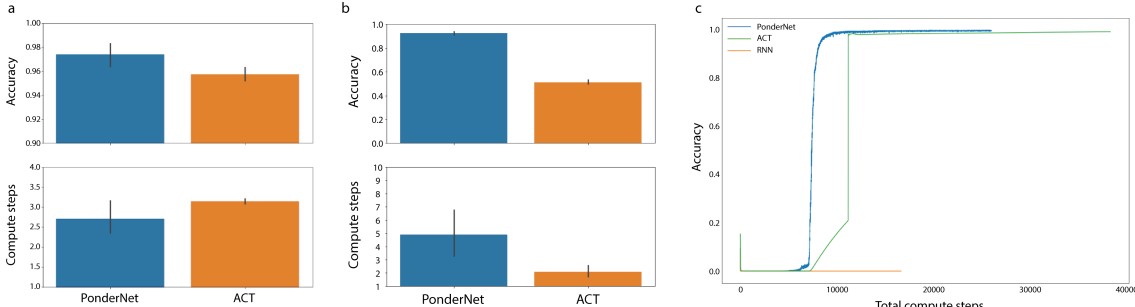

Figure 1: Performance on the parity task. a) Interpolation. Top: accuracy for both PonderNet(blue) and ACT(orange). Bottom: number of ponder steps at evaluation time. Error bars calculated over 10 random seeds. b) Extrapolation. Top: accuracy for both PonderNet(blue) and ACT(orange). Bottom: number of ponder steps at evaluation time. Error bars calculated over 10 random seeds. c) Total number of compute steps calculated as the number of actual forward passes performed by each network. Blue is PonderNet, Green is ACT and Orange is an RNN without adaptive compute.

our method is particularly robust with respect to the prior, and a clear advancement in comparison to ACT, where the $\tau$ parameter is difficult to set and it is a source of training instability, as explained in the original paper and confirmed by our results. Indeed, Fig. 2a shows that only for few configuration of $\tau$ ACT is able to solve the task and even when it does so there is a great variance across seeds. Finally, one advantage of setting a prior probability is that this parameter is easy to interpret as the inverse of the "number of ponder steps", whereas the $\tau$ parameter does not have any straightforward interpretation, which makes it harder to define a priori.

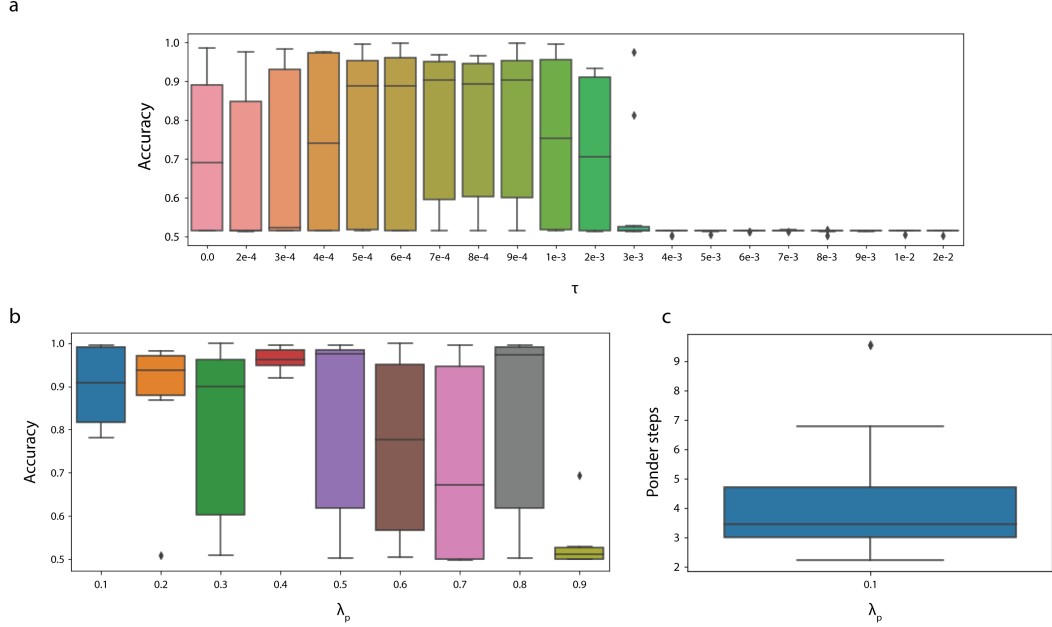

Figure 2: Sensitivity to hyper-parameter. a) Sensitivity of ACT to $\tau$. Each box-plot is over 10 random seeds. b) Sensitivity of PonderNet to $\lambda_p$. Each box-plot is over 10 random seeds. c) Box-plot over 30 random seeds for number of ponder steps when $\lambda_p = 0.1$.

Next we moved to test the ability of PonderNet to allow extrapolation. To do this we consider input vectors of 96 elements instead. We train the network on input vectors up from integers ranging from 1 to 48 elements and we then evaluate on integers between 49 and 96. Figure 1b shows that PonderNet was able to achieve almost perfect accuracy on this hard extrapolation task, whereas ACT remained at chance level. It is interesting to see how PonderNet increased its thinking time to 5 steps, which is almost twice as much as the ones in the interpolation set (see Fig. 1a), showing the capability of our method to adapt its computation to the complexity of the task.

## 3.2 bAbI

We then turn our attention to the bAbI question answering dataset (Weston et al., 2015), which consists of 20 different tasks. This task was chosen as it proved to be difficult for standard neural network architecture that do not employ adaptive computation (Dehghani et al., 2018). In particular we trained our model on the joint 10k training set. Also, please see Appendix C for further training and evaluation details.

Table 1 reports the averaged accuracy of our model and the other baselines on bAbI. Our model is able to match state of the art results, but it achieves them faster and with a lower average error. The comparison with Universal transformer (Dehghani et al., 2018, UT) is interesting as it uses the same transformer architecture as PonderNet, but the compute time is optimised with ACT. Interestingly, to solve 20 tasks, Universal Transformer takes 10161 steps, whereas our methods 1658, hence confirming that approach uses less compute than ACT.

| Architecture | Average Error | Tasks Solved |
|---|---|---|
| Memory Networks (Sukhbaatar et al., 2015) | 4.2± 0.2 | 17 |
| DNC (Graves, 2016) | 3.8± 0.6 | 18 |
| Universal Transformer (Dehghani et al., 2018) | 0.29± 1.4 | 20 |
| Transformer+PonderNet | 0.15± 0.9 | 20 |

Table 1: bAbI. Test results chosen by validation loss. Average error is calculated over 5 seeds

## 3.3 Paired associative inference

Finally, we tested PonderNet on the Paired associative inference task (PAI) (Banino et al., 2020). This task is thought to capture the essence of reasoning – the appreciation of distant relationships among elements distributed across multiple facts or memories and it has been shown to benefit from the addition of adaptive computation. Please refer to Appendix D for further details on the task and the training regime.

Table 2 reports the averaged accuracy of our model and the other baselines on PAI. Our model is able to match the results of MEMO, which was specifically designed with this task in mind. More interestingly, our model although is using the same architecture as UT (Dehghani et al., 2018) is able to achieve higher accuracy. For the complete set of results please see Table 7 in Appendix D.

| Length | UT | MEMO | PonderNet |
|---|---|---|---|
| 3 items (trained on: A-B-C - accuracy on A-C) | 85.60 | 98.26(0.67) | 97.86(3.78) |

Table 2: Inference trial accuracy. PonderNet results chosen by validation loss, averaged on 3 seeds. For Universal Transformer (UT) and MEMO the results were taken from Banino et al. (2020)

## 4. Discussion

We introduced PonderNet, a new algorithm for learning to adapt the computational complexity of neural networks. It optimizes a novel objective function that combines prediction accuracy with a regularization term that incentivizes exploration over the pondering time. We demonstrated on the parity task that a neural network equipped with PonderNet can increase its computation to extrapolate beyond the data seen during training. Also, we showed that our methods achieved the highest accuracy in complex domains such as question answering and multi-step reasoning. Finally, adapting existing recurrent architectures to work with PonderNet is very easy: it simply requires to augment the step function with an additional halting unit, and to add an extra term to the loss. Critically, we showed that this extra loss term is robust to the choice of $\lambda_p$, the hyper-parameter that defines a prior on how likely is that the network will halt, which is an important advancement over ACT.

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

## Appendix A. Comparison to ACT

PonderNet builds on the ideas introduced in Adaptive Computation Time (ACT; Graves, 2016). The main contribution of this paper is to reformulate how the network learns to halt in a probabilistic way. This has consequences in all aspects of the model, including: the architecture and forward computation; the loss used to train the network; the deployment of the model; and the limitation of how multiple pondering modules can be combined. We explain in more detail all these differences below.

### A.1 Forward computation

PonderNet's step function (that is computed on every step) is identical to the one proposed in ACT. They both assume a mapping $y_n, h_{n+1}, \lambda_n = s(x, h_n)$. The main difference between ACT and PonderNet's forward computation is how the halting node $\lambda_n$ is used.

In ACT, the network is unrolled for a number of steps $N_{ACT} = min\{N : \sum_{n=1}^{N} \lambda_n \geq 1 - \epsilon\}$. ACT's halting nodes learn to predict the overall probability that the network halted at step $n$, so that $\lambda_n = p_n$. The value of the halting node in the last step is replaced with a *remainder* quantity $R = \lambda_N = p_N = 1 - \sum_{n=1}^{N-1} \lambda_n$. In ACT it would not make sense to unroll the network for a larger number of steps than $N_{ACT}$ because the sum of probabilities of halting would be $> 1$. When training ACT, higher values of $N$ are not necessarily better, and $N$ is being determined (and learnt) via the halting node $\lambda_n$. In PonderNet, any sufficiently high value of $N$ can be used, and the unroll length of the network at training is distinguished from the learning of the halting policy (which is most critical for saving computation when deployed at evaluation).

The output of ACT is not treated probabilistically but as a weighted average $\hat{y}_{ACT} = \sum_{n=1}^{N_{ACT}} \hat{y}_n \lambda_n$ over the outputs at each step. The halting, as well as the output, are computed identically for training and evaluation. In PonderNet, the output is probabilistic. In training, we compute the output and halting probabilities across many steps so that we can compute a weighted average of the *loss*. In evaluation, the network returns its prediction as soon as a halt state is sampled.

Finally, ACT considers the case of sequential data, where the step function can ponder dynamically for each new item in the input sequence. Given the introduction of attention mechanisms in the recent years (e.g. Transformers; Vaswani et al., 2017) that can process arrays with dynamic shapes, we suggest that pondering should be done holistically instead of independently for each item in the sequence. This can be useful in learning e.g. how many message-passing steps to do in a graph network (Veličković et al., 2019).

### A.2 Training loss

ACT proposes a heuristic training loss that combines two intuitive costs: the accuracy of the model, and the cost of computation. These two costs are in different units, and not easily comparable. Since $N_{ACT}$ is not differentiable with respect to $\lambda_n$, ACT utilizes the remainder $R = 1 - \sum_{n=1}^{N-1}$ as a proxy for minimizing the total number of computational steps. This is unlike in PonderNet, where the expected number of steps can be computed (and differentiated) exactly as $\sum_{n=1}^{N} np_n$.

In PonderNet, however, we propose that naively minimizing the number of steps (subject to good performance) is not necessarily a good objective. Instead, we propose that matching a prior halting distribution has multiple benefits: a) it provides an incentive for exploring alternative halting strategies; b) it provides robustness of the learnt step function, which may improve generalization; c) the KL is in same units as information-theoretic losses such as cross-entropy; and d) it provides an incentive to not ponder for longer than the prior.

Note that in PonderNet, we compute the loss for every possible number of computational steps, and then minimize the expectation (weighted average) over those. This is unlike in ACT where the expectation is taken over the predictions, and a loss is computed by comparing the average prediction with the target. This has the consequence that combining multiple networks is easier in ACT than in PonderNet. One could easily chain multiple ACT modules next to each other, and the size of the network during training would grow linearly with the number of modules. However, the network size when chaining PonderNet modules grows exponentially because the loss would need to be estimated conditioned on each PonderNet module halting at each step.

In PonderNet we have introduced two loss hyper-parameters $\lambda_p$ and $\beta$, in comparison to a single hyper-parameter $\tau$ in ACT that trades-off accuracy with computational complexity. We note that, while $\tau$ and $\beta$ are superficially similar (they both apply a weight to the regularization term), their effect is not equivalent because the regularization of ACT and PonderNet have different interpretation.

### A.3 Evaluation

ACT's predictions are computed identically during training and evaluation. In both contexts, the maximum number of steps $N_{ACT}$ is determined based on the inputs, and the prediction is computed as a weighted average over the predictions in all steps. In PonderNet, training and evaluation are performed differently. During evaluation, the network halts probabilistically by sampling $\Lambda_n$, and either outputs the current prediction or performs an additional computational step. During training, we are not interested in the predictions per se but in the expected loss over steps, and so estimate this up to a maximum number of steps $N$ (the higher the better). This estimate will improve with higher probability that the network has halted at some point during the first $N$ steps (i.e. the cumulative probability of halting).

## Appendix B. Parity.

### B.1 Training and evaluation details

For this experiment we used the Parity task as explained by Graves (2016).

All the models used the same architecture, a simple RNN with a single hidden layer containing 128 tanh units and a single logistic sigmoid output unit. All models were optimized using Adam (Kingma and Ba, 2014), with learning rate fixed to 0.0003. The networks were trained with binary cross-entropy loss to predict the corresponding target, 1 if there was an odd number of ones and 0 if there was an even number of ones. We used minibatches of size 128. For both architectures the weights were optimised using Adam (Kingma and Ba, 2014), with learning rate fixed to 0.0003. For PonderNet we sampled uniformly 10 values

of $\lambda_p$ in the range (0, 1]. For ACT we sampled uniformly 19 values of $\tau$ in the range [2e-4, 2e-2] and we added also 0, which correspond to not penalising the halting unit at all. For both ACT and Ponder, $N$ was set to 20. For PonderNet $\beta$ was fixed to 0.01

## Appendix C. bAbI.

### C.1 Training and evaluation details

For this experiment we used the English Question Answer dataset Weston et al. (2015). We use the training and test datasets that they provide with the following pre-processing:

- All text is converted to lowercase.

- Periods and interrogation marks were ignored.

- Blank spaces are taken as word separation tokens.

- Commas only appear in answers, and they are *not* ignored. This means that, e.g. for the path finding task, the answer 'n,s' has its own independent label from the answer 'n,w'. This also implies that every input (consisting of 'query' and 'stories') corresponds to a single answer throughout the whole dataset.

- All the questions are stripped out from the text and put separately (given as "queries" to our system).

At training time, we sample a mini-batch of 64 queries from the training dataset, as well as its corresponding stories (which consist of the text prior to the question). As a result, the queries are a matrix of $128 \times 11$ tokens, and sentences are of size $128 \times 320 \times 11$, where 128 is the batch size, 320 is the max number of stories, and 11 is the max sentence size. We pad with zeros every query and group of stories that do not reach the max sentence and stories size. For PonderNet, stories and query are used as their naturally corresponding inputs in their architecture. The details of the network architecture are described in Section C.2.

After that mini-batch is sampled, we perform one optimization step using Adam Kingma and Ba (2014). We also performed a search on hyperparameters to train on bAbI, with ranges reported on Table 4. The network was trained for $2e4$ epochs, each one formed by 100 batch updates.

For evaluation, we sample a batch of $10,000$ elements from the dataset and compute the forward pass in the same fashion as done in training. With that, we compute the mean accuracy over those examples, as well as the accuracy per task for each of the 20 tasks of bAbI. We report average values and standard deviation over the best 5 hyper parameters we used.

For MEMO the results were taken from Banino et al. (2020) and for Universal transformer we used the results in Dehghani et al. (2018).

### C.2 Transformer architecture and hyperparameters

We use the same architecture as described in Dehghani et al. (2018). More concretely, we use the implementation and hyperparameters described as 'universal_transformer_small' that is

available at `https://bit.ly/3frofUI`. For completeness, we describe the hyperparameters used on Table 3.

We also performed a search on hyperparameters to train on our tasks, with ranges reported on Table 4.

| Parameter name | Value |
|---|---|
| Optimizer algorithm | Adam |
| Learning rate | 3e-4 |
| Input embedding size | 128 |
| Attention type | as in Vaswani et al. (2017) |
| Attention hidden size | 512 |
| Attention number of heads | 8 |
| Transition function | MLP(1 Layer) |
| Transition hidden size | 128 |
| Attention dropout rate | 0.1 |
| Activation function | RELU |
| N | 10 |
| $\beta$ | 0.01 |

Table 3: Hyperparameters used for bAbI experiments.

| Parameter name | Value |
|---|---|
| Attention hidden size | {128, 512} |
| Transition hidden size | {128, 512} |
| $\lambda_p$ | uniform(0, 1.0] |

Table 4: Hyperparameters ranges used to search over with PonderNet on bAbI.

## Appendix D. Paired Associative Inference

### D.1 PAI - Task details

For this task we used the dataset published in Banino et al. (2020), also the task is available at `https://github.com/deepmind/deepmind-research/tree/master/memo`

To build the dataset, Banino et al. (2020) started with raw images from the ImageNet dataset (Deng et al., 2009), which were embedded using a pre-trained ResNet (He et al., 2016), resulting in embeddings of size 1000. Here we are focusing on the dataset with sequences of length three (i.e. $A - B - C$) items, which is composed of $1e6$ training images, $1e5$ evaluation images and $2e5$ testing images.

A single entry in the batch is built by selecting $N = 16$ sequences from the relevant pool (e.g. training) and it's composed by three items:

- a memory,

- a query,

- a target.

Each memory content is created by storing all the possible pair wise association between the items in the sequence, e.g. $A_1B_1$ and $B_1C_1$, $A_2B_2$ and $B_2C_2$, ..., $A_NB_N$ and $B_NC_N$. With $N = 16$, this process results in a memory with $M = 32$ rows each one with 2 embeddings of size 1000.
Each query is composed of 3 images, namely:

- the cue

- the match

- the lure

The cue (e.g. $A_1$) and the match (e.g. $C_1$) are images extracted from the sequence; whereas the lure is an image from the same memory content but from a different sequence (e.g. $C_7$). There are two types of queries - "direct" and "indirect". In "direct" queries the cue and the match are sampled from the same memory slot. For example, if the sequence is $A_1$ - $B_1$ - $C_1$, then an example of direct query would be, $A_1$ (cue) - $B_1$ (match) - $B_{12}$ (lure). More of interests here is the case of "indirect" queries, as they require an inference across multiple facts stored at different location in memory. For instance, if we consider again the previous example sequence: $A_1$ - $B_1$ - $C_1$, then an example of inference trial would be $A_1$ (cue) - $C_1$ (match) - $C_6$ (lure).

The queries are presented to the network as a concatenation of three image embedding vectors (the cue, the match and the lure), that is a $3 \times 1000$ dimensional vector. The cue is always placed in the first position in the concatenation, but to avoid any trivial solution, the position of the match and lure are randomized. It is worth noting that the lure image always has the same position in the sequence (e.g. if the match image is a C the lure is also a C) but it is randomly drawn from a different sequence that is also present in the current memory. This way the task can only be solved by appreciating the correct connection between the images, and this need to be done by avoiding the interference coming for other items in memory. For each entry in the batch we generated all possible queries that the current memory store could support and then one was selected at random. Finally, the batch was balanced, i.e. half of the elements were direct queries and the other half was indirect. Finally, the targets represent the ImageNet class-ID of the matches.
To summarize, for each entry in each batch:

- Memory was of size $32 * 2 * 1000$

- Queries were of size $1 * 3 * 1000$

- Target was of size 1

## D.2 PAI - Architecture details

We used an architecture similar to Universal Transformers (Dehghani et al., 2018, UT), but we augmented the transformer with a memory as in Dai et al. (2019). The number of layers in the encoder and the decoder was learnt, but constrained to be the same. This number was identified as the "pondering time" in our PonderNet architecture. Also, we set an upper bound $N$ to the number of layers. The initial state $h_0$ was a learnt embedding of

the input. On each step, the state was updated by applying the encoder layer once, that is: $h_{n+1} = encoder(h_n)$. Note that in this case PonderNet only received information about the inputs through its state. The prediction was computed by applying the decoder layer an equal number of times to the pondering step, that is $\hat{y}_{n+1} = decoder(...(decoder(h_{n+1}))$. With this architecture, PonderNet was able to optimize how many times to apply the encoder and the decoder layers to improve its performance in this task.

The weights were optimised using Adam (Kingma and Ba, 2014), using polynomial weight decay with a maximum learning rate of 0.0003 and learning rate linear warm-up for the first epoch. The mini-batch size was of size 128. For completeness, we describe the hyperparameters used on Table 5. We also performed a search on hyperparameters to train on our tasks, with ranges reported on Table 6.

| Parameter name | Value |
|---|---|
| Optimizer algorithm | Adam |
| Input embedding size | 256 |
| Attention type | as in Vaswani et al. (2017) |
| Attention hidden size | 512 |
| Attention number of heads | 8 |
| Transition function | MLP(2 Layers) |
| Transition hidden size | 128 |
| Attention dropout rate | 0.1 |
| $\beta$ | 0.01 |

Table 5: Hyperparameters used for PAI experiments.

| Parameter name | Value |
|---|---|
| Attention hidden size | {256, 512} |
| Transition hidden size | {128, 1024} |
| $\lambda_p$ | uniform(0, 0.5] |
| N | [7, 10] |

Table 6: Hyperparameters ranges used to search over with PonderNet on PAI.

### D.3 PAI - Results based on query type

The result reported below in Table 7 are from the evaluation set at the end of training. Each evaluation set contains 600 items.

Table 7: Paired Associative - length 3: A-B-C

| Trial Type | MEMO | UT | PonderNet |
|---|---|---|---|
| A-B | 99.82(0.30) | 97.43 | 98.01(2.39) |
| B-C | 99.76(0.38) | 98.28 | 97.43(1.97) |
| A-C | 98.26(0.67) | 85.60 | 97.86(3.78) |

For MEMO and for Universal transformer the results were taken from Banino et al. (2020).

## Appendix E. Broader impact statement

In this work we introduced PonderNet, a new method that enables neural networks to adapt their computational complexity to the task they are trying to solve. Neural networks achieve state of the art in a wide range of applications, including natural language processing, reinforcement learning, computer vision and more. Currently, they require much time, expensive hardware and energy to train and to deploy. They also often fail to generalize and to extrapolate to conditions beyond their training.

PonderNet expands the capabilities of neural networks, by letting them decide to ponder for an indefinite amount of time (analogous to how both humans and computers think). This can be used to reduce the amount of compute and energy at inference time, which makes it particularly well suited for platforms with limited resources such as mobile phones. Additionally, our experiments show that enabling neural networks to adapt their computational complexity has also benefits for their performance (beyond the computational requirements) when evaluating outside of the training distribution, which is one of the limiting factors when applying neural networks for real-world problems.

We encourage other researchers to pursue the questions we have considered on this work. We believe that biasing neural network architectures to behave more like algorithms, and less like "flat" mappings, will help develop deep learning methods to their the full potential.

