# OpenReview forum: "PonderNet: Learning to Ponder"
_ICML.cc/2021/Workshop/AutoML — AutoML@ICML2021 Poster_

### Official Review · Reviewer_s6dq · 2021-06-13
**Comments**

**Rating:** 6
**Confidence:** 3

**Review:**

In practice, while using the state of the art DNN architectures, the computational complexity grows with the size of the input as opposed to the complexity of the downstream task. To overcome the above limitation the authors proposed PonderNet, which learns to adapt the computation depending on the complexity of the task without compromising much on the accuracy & the generalisation. This paper explores a promising direction where the model automatically adjusts its computation based on the complexity of the task, this would, in turn, be efficient in practice as opposed to picking up off-the-shelf architectures which are in most cases an overkill for the problem in hand.

---

### Official Review · Reviewer_oWVZ · 2021-06-14
**Interesting contribution, although the impact is yet unclear**

**Rating:** 6
**Confidence:** 4

**Review:**

The paper proposes a novel method that halts the number of computational steps in order to trade-off between accuracy and computation cost. The method is an incremental contribution on top of the Adaptive Computation Time approach (Graves, 2016) with a novel focus on handling its stability. Concretely, the paper intriduces a different halting process through a halting node mechanism, as well as a combined loss function with a reconstruction and a regularization term.

Overall, the idea is convincing and principled, eventhough the experiments are not convincingly shown for a wide range of popular deep learning datasets, e.g. typical computer vision classification problems are missing, which makes it hard to judge the impact of the method.

Another key question is how efficiently can the halting mechanism and the sequential predictions for different n be integrated with the state-of-the-art deep learning models on relevant tasks, e.g. adding the concept of conditional halting to the layers of EfficientNet on Imagenet. I feel the paper falls short in that aspect.

Last, but not least, the formalism demands some polishing, e.g. the term p_G(\lambda_p) at the loss of Equation 3 has not been previously defined.

---

### Decision · Program_Chairs · 2021-06-21

Accept (Poster)